

# Fabrication of calixarene-grafted bio-polymeric magnetic composites for magnetic solid phase extraction of non-steroidal anti-inflammatory drugs in water samples

Syed Fariq Fathullah Syed Yaacob[1], Arniza Khairani Mohd Jamil[1], Muhammad Afzal Kamboh[2], Wan Aini Wan Ibrahim[3] and Sharifah Mohamad[1,4]

[1] Department of Chemistry, Faculty of Science, Universiti Malaya, Kuala Lumpur, Malaysia
[2] Department of Chemistry, Shaheed Benazir Bhutto University, Shaheed Benazirabad, Sindh, Pakistan
[3] Separation Science and Technology Group (SepSTec), Department of Chemistry, Faculty of Science, Universiti Teknologi Malaysia, Johor Bahru, Johor, Malaysia
[4] Universiti Malaya Centre for Ionic Liquids, Universiti Malaya, Kuala Lumpur, Malaysia

Corresponding author
Sharifah Mohamad,
sharifahm@um.edu.my

## ABSTRACT

Calixarene framework functionalized bio-polymeric magnetic composites (MSp-TDI-calix) were synthesized and utilized as magnetic solid-phase extraction (MSPE) sorbent for the extraction of non-steroidal anti-inflammatory drugs (NSAIDs), namely indoprofen (INP), ketoprofen (KTP), ibuprofen (IBP) and fenoprofen (FNP), from environmental water samples. MSp-TDI-calix was characterized by FT-IR, XRD, FESEM, EDX, VSM and BET analysis, and the results were compared with Sp-TDI and Sp-TDI-calix. To maximize the extraction performance of MSp-TDI-calix decisive MSPE affective parameters such as sorbent amount, extraction time, sample volume, type of organic eluent, volume of organic eluent, desorption time and pH were comprehensively optimized prior to HPLC-DAD determination. The analytical validity of the proposed MSPE method was evaluated under optimized conditions and the following figures of merit were acquired: linearity with good determination coefficient ($R^2 \geq 0.991$) over the concentration range of 0.5–500 µg/L, limits of detection (LODs) ranged from 0.06–0.26 µg/L and limits of quantitation (LOQ) between 0.20–0.89 µg/L. Excellent reproducibility and repeatability under harsh environment with inter-day and intra-day relative standard deviations were obtained in the range of 2.5–3.2% and 2.4–3.9% respectively. The proposed method was successfully applied for analysis of NSAIDs in tap water, drinking water and river water with recovery efficiency ranging from 88.1–115.8% with %RSD of 1.6–4.6%.

## INTRODUCTION

From an environmental point of view, the mixing of pharmaceuticals as well as their degraded products into natural streams is of great concern (*Lacey et al., 2008*; *Toledo-Neira & Álvarez Lueje, 2015*). Recent reports indicated that drugs from pharmaceutical waste have long-term adverse effects to humans and the aquatic system (*Van Hoeck et al., 2009*; *Aguilar-Arteaga et al., 2010*). Non-steroidal anti-inflammatory drugs (NSAIDs) is one of the most important groups of pharmaceutical drugs to treat pain and inflammation. These compounds inhibit the production of cyclooxygenase enzymes and reduce the prostaglandins level that causes pain, inflammation and fever within the human body (*Fan et al., 2014*; *Amiri et al., 2016*). Overdose of NSAIDs generally cause adverse side effects such as ulcers, kidney failure and stomach bleeding which can ultimately lead to sudden death. These side effects are more prominent for those with poor health conditions and alcoholics (*Shukri et al., 2015*). The NSAIDs may transfer to natural streams through hospital and pharmaceutical units, effluent, as well as private household manure (*Toledo-Neira & Álvarez Lueje, 2015*). These drugs pose toxic effects in aquatic ecosystems and may cause harm not only to marine life but also to humans (*Aguilar-Arteaga et al., 2010*). Consequently, the precise determination of NSAIDs is of prime importance. Pharmaceutical compounds are mainly present in water samples at trace levels (*Luo et al., 2014*). It is a challenge to reach down to this detection level due to complicated matrix effects in real samples and thus this needs to be adequately addressed. Several analytical techniques such as gas chromatography (GC) (*Samaras et al., 2011*), capillary electrophoresis (CE) (*Macià et al., 2004*), liquid chromatography mass spectrometry (LC-MS) (*Toledo-Neira & Álvarez Lueje, 2015*) and high performance liquid chromatography (HPLC) (*Aguilar-Arteaga et al., 2010*) have been employed for NSAIDs determination. Comparatively, due to high resolution, great automation and high reproducibility, HPLC is considered as one of the most commonly used techniques (*Fan et al., 2014*).

Various sample preparation methods have been reported for the enrichment and purification of pharmaceutical compounds. Liquid-liquid extraction (LLE) (*Wen, Tu & Lee, 2004*; *Payán et al., 2009*), solid phase extraction (SPE) (*Santos et al., 2005*; *Rodil et al., 2009*), stir rod sorptive extraction (SRSE) (*Luo et al., 2011*), dispersive liquid-liquid microextraction (DLLME) (*Shukri et al., 2015*), and solid phase micro-extraction (SPME) (*Moeder et al., 2000*; *Torres Padrón, Sosa Ferrera & Santana Rodríguez, 2009*) methods are commonly used for determination of drugs and their metabolites. Among these techniques, solid phase extraction is one of the most virtual and reliable methods due to its advantages of producing high enrichment factors, magnificent selectivity and high recovery (*Li & Lee, 2001*). However, the classical SPE method has disadvantages such as solvent loss, time consumption, large secondary waste, expensive complex equipment and tedious procedure (*Asgharinezhad et al., 2014*). Magnetic solid phase extraction (MSPE) was developed as an alternative method to the SPE procedure due to its spectacular characteristics which can overcome the limitations of SPE. Simple procedure, economical, high efficiency, rapidity,

less consumption of organic solvent and no filtration and/or centrifugation required for sample preparation are some benefits of MSPE (*Wang et al., 2013*; *Wan Ibrahim et al., 2015*).

To acquire adequate recovery of target analytes, appropriate sorbent selection is essential in MSPE procedures. Iron oxides such as magnetite ($Fe_3O_4$) are an example of magnetic nanoparticles (MNPs) that are widely used as a sorbent because of their high surface area, economical value, low toxicity, simple preparation steps and easy separation of target analytes from samples (*Rossi, Quach & Rosenzweig, 2004*; *Tahmasebi & Yamini, 2012*). The use of MNPs significantly decrease the extraction period to enable the balance between solid phase and analytes and also to reduce the consumption of sorbent mass even with large volumes of samples (*Wierucka & Biziuk, 2014*). However, naked MNPs are instable in acidic medium, easily oxidize, agglomerate in aqueous solution and lack selectivity (*Xu et al., 2014*). Thus, various approaches can be adopted to modify the surface of MNPs via functionalization or immobilization in order to overcome these limitations. One such approach is the modification of MNPs with a bio-polymeric material known as sporopollenin as a potential robust sorbent (*Kaya et al., 2016*). Sporopollenin forms the tough spore/pollen wall of a land plant known as "*Lycopodium clavatum*". The aromatic rings of sporopollenin contain carbon, hydrogen and oxygen with the stoichiometry formula of $C_{90}H_{144}O_{27}$ (*Kamboh et al., 2016*). The superior properties of sporopollenin are its resistance towards harsh chemicals, high thermal stability, stability against mineral acid and alkaline and possession of excellent physical and chemical strength (*Crini & Ndongo Peindy, 2006*; *Ayar, Gezici & Küçükosmanoğlu, 2007*). Apart from that, sporopollenin is also economical, ecological and readily accessible (*Barrier et al., 2011*; *Mundargi et al., 2016*). Thus, sporopollenin is a potential bio-polymer material sorbent for MNPs modification by functionalizing the sporopollenin surface with suitable functional groups for extraction of target analytes. Furthermore, the inner and outer surface of sporopollenin is available for binding with guest molecule due to its 2 $\mu$m thick perforated walls hollowed exine which makes it porous (*Kamboh et al., 2016*). The MNPs can easily embed on the surface of sporopollenin due to the availability of a large internal cavity of exine, allowing it to be easily magnetized by MNPs.

Calixarenes is a remarkable macrocyclic host compound that can be synthesized by the oligomerization of phenol and formaldehyde (*Demirkol et al., 2014*; *Gokoglan et al., 2015*). The outstanding properties possessed by calixarenes make its inner cavity surface hydrophobic as well as its outer surface hydrophilic which can be tuned structurally to achieve desired properties (*Kamboh et al., 2016*). Calixarenes can form a host guest inclusion complex with a variety of guest compounds such as ions, amino acid and metal ions by functionalization of the upper or lower rim of the molecule. Hence, calixarenes can be extensively used as a trapping agent to encapsulate target compounds through host-guest interactions. By combining calixarene modified magnetic nanoparticles and sporopollenin as solid support, herein we introduce the novel nanocomposite for extraction of NSAIDs from water samples. Based on reported literature, the preparation of calixarene modified magnetic nanoparticles mostly involves tedious and lengthy preparation procedures ($\sim$40 days of reflux) (*Gubbuk et al., 2012*; *Sayin, Ozcan & Yilmaz, 2013*; *Kamboh et al., 2016*).

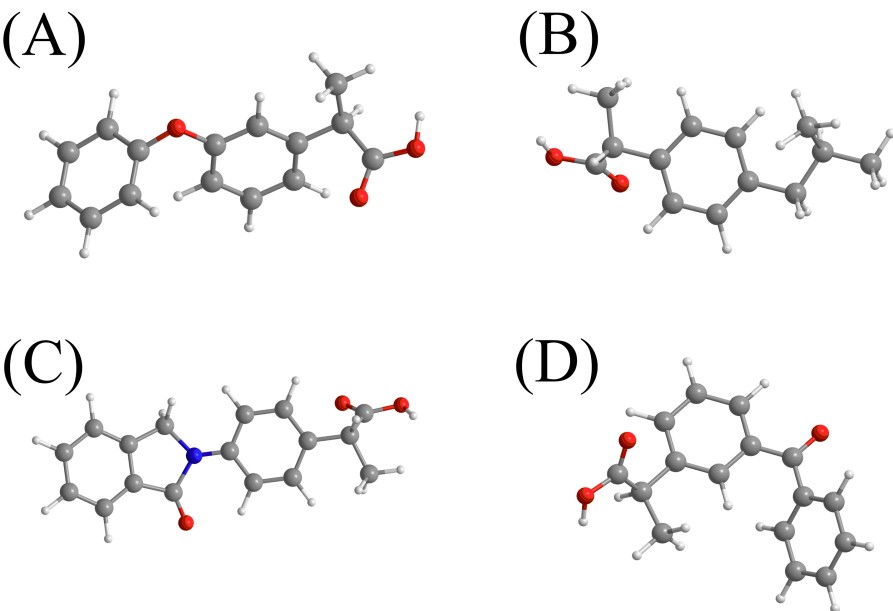

**Figure 1** **The chemical structures of the selected NSAIDs and pK$_a$-value.** (A) fenoprofen (pK$_a$ = 4.50)[b]; (B) ibuprofen (pK$_a$ = 4.51)[a]; (C) indoprofen (pK$_a$ = 4.40)[a]; (D) ketoprofen (pK$_a$ = 4.20)[a]. [a]Ref (*Cha & Myung, 2013*). [b]http://www.drugbank.ca (accessed on 6 th July 2017).

Hence, we aim to initiate a simpler and shorter (<10 h) synthesis procedure and easier preparation of the sorbent. Other than that, using less organic solvent for preparation of the calixarene modified magnetic nanoparticles compared to reported publications would be a crucial factor in preserving the environment towards a greener research (*Sayin & Yilmaz, 2011*).

In this study, a bio-polymeric material sorbent modified with *p*-tert-butylcalix[4]arene was employed to extract and detect NSAIDs in water samples. The sorbent surface was also magnetized with MNPs to ease separation using an external magnetic field. The role of host guest encapsulation of calixarene makes it a suitable host candidate to entrap and extract the NSAIDs from environmental samples by applying the MSPE technique. In addition, the hydrogen bonding, $\pi-\pi$ interaction and hydrophobic interaction that could be formed with NSAIDs would allow the modified nanocomposite to selectively entrap and bind the NSAIDs at the surface. The NSAIDs group namely INP, KTP, IBP and FNP as illustrated in Fig. 1 were selected as analytes because of their frequent consumption as drugs in Malaysia based on an Anatomical Therapeutic Chemical (ATC) report (*Khairudin et al., 2017*). Hence, the occurrence probability of these NSAIDs in water samples is very high due to their high consumption among the Malaysian population (*Khairudin et al., 2017*).

## MATERIALS AND METHODS

### Materials

*Lycopodium clavatum* sporopollenin (25 μm particle size), toluene-2,4-diisocyanate (TDI) (CAS no 584-84-9) was purchased from Sigma-Aldrich (Steinheim, Germany). Indoprofen

(CAS no. 34842-01-0), ibuprofen (CAS no. 15687-27-1) and fenoprofen (CAS no. 53746-45-5) were purchased from Sigma-Aldrich (Steinheim, Germany). Ketoprofen (CAT no. 155154) was purchased from MP Biomedicals Inc. (Solon, Ohio, USA). HPLC-grade acetonitrile was provided by Merck (Dermstadt, Germany) and HPLC-grade acetic acid was provided by J.T Baker Avantor Performance Material Inc. (Center Valley, PA, USA). Analytical grade ammonia, dichloromethane, ethanol, *n*-hexane, toluene, chloroform, ethyl acetate, dimethylformamide, hydrochloric acid and sodium hydroxide pellet were purchased from Merck (Darmstadt, Germany) and used without further purification. Dry toluene and dry N,N-dimethylformamide (DMF) were obtained from Merck (Dermstadt, Germany). Ferrous chloride tetrahydrate (CAS no. 13478-10-9) and ferric chloride hexahydrate (CAS no. 10025-77-1) were obtained from Sigma-Aldrich (Steinheim, Germany). Analytical grade methanol used for the preparation of standard working solutions was obtained from Merck (Dermstadt, Germany). The pH of the solution was adjusted by mixing appropriate volumes of 0.1 M hydrochloric acid (HCl) and/or sodium hydroxide (NaOH). Deionized water that had been passed through a Milli-Q system (Lane End, UK) was used during the preparation of solutions. Deionized water was filtered through a 0.22 µm nylon filter (Whatman, England) for preparing the mobile phase solvent before adding 1% acetic acid. The *p*-tert-butylcalix[4]arene was synthesized according to a previously reported method (*Gutsche, Iqbal & Stewart, 1986*).

## Standard solutions

Stock solutions of NSAIDs were prepared in methanol and stored at 2 °C in the refrigerator. The stock solutions were prepared fresh monthly due to their stability. The working solutions were prepared daily by dilution from stock solution with a final concentration of 1 mg/L using the same solvent as the stock solution. Calibration curve ($n = 7$) was prepared by serial dilution of the mixed working solution in deionized water and performing MSPE procedure as described below. The resulting individual concentrations ranged from 0.5–500 µg/L (0.5, 1, 5, 10, 50, 100, 500 µg/L). The samples were injected into the HPLC system under optimized conditions and the chromatogram was recorded at multiple wavelengths respectively. Each measurement was done in triplicates.

## Quantification and method validation

Method validation was carried out in terms of linearity, precision, limit of detection (LOD), and limit of quantification (LOQ). The linearity of the method was measured at seven concentrations from 0.5 to 500 µg/L as described earlier. The calibration curves were plotted by peak area as the *y*-axis versus concentration as the *x*-axis and the linear regression equations were calculated for each analyte. The precision was expressed in relative standard deviation (RSD) and evaluated in terms of intra-day (repeatability) and inter-day (reproducibility). Intra-day was assessed by performing five replicate analyses on a single day ($n = 5$) and inter-day (reproducibility) was carried out by performing five replicates on three consecutive days ($n = 15$). LOD was determined based on the signal-to-noise ratio and is defined mathematically as equal to three times the standard deviation of the blank (3SD/$m$) and LOQ is 10 times the standard deviation of the blank

(10SD/$m$) where SD is the standard deviation from 10 injections of blank samples ($n = 10$) and $m$ is the gradient slope from the calibration curve.

## Instruments

The Fourier transform-infrared spectroscopy (FT-IR) spectrum was obtained using ATR mode on a Spectrum 400 Perkin Elmer (Waltham, MA, USA) in the range of 4,000–450 cm$^{-1}$ with diamond as the detector. X-ray diffraction (XRD) patterns of the samples were taken using Panalytical Empyrean X-ray diffractometer (EA Almelo, The Netherlands) from $2\Theta = 15°$ to $75°$ f at room temperature utilizing Cu K$\alpha$ radiation at a wavelength of 1.5418 Å at a scan rate of 0.02 s$^{-1}$. Field emission scanning electron microscopy with energy dispersive X-ray spectroscopy (FESEM-EDX) analysis was performed by using the Hitachi SU8220 scanning electronic microscopy from Oxford Instruments (Oxfordshire, UK). The magnetization of functionalized MNP was measured using a vibrating sample magnetometer (VSM LakeShore 7400 series, Carson, CA, USA). The surface area and porosity were measured using Brunauer-Emmett-Teller (BET) by nitrogen adsorption–desorption isotherm in Micromeritics Tristar II ASAP 2020, (Norcross, GA, USA). The HPLC system (Kyoto, Japan) based on LC-20AT pump, SPD-M20A diode array detector, SIL-20A HT autosampler and CTO-10AS VP column oven was used for NSAIDs determination. The system was equipped with a Hypersil gold C-18 reverse phase column (250 × 4.6 mm), particle size (5 μm) from ThermoScience (Waltham, MA, USA).

## HPLC parameters and conditions

The HPLC-DAD system was used for the chromatographic identification of selected NSAIDs from the water samples. Chromatographic separation of the selected NSAIDs were carried out using acidified (1% with acetic acid) water/acetonitrile (50:50 v/v) as the mobile phase at a flow rate of 1.0 mL min$^{-1}$. The HPLC column temperature was set at 40 °C. The sample injection volume was 10 μL. The DAD detection for the selected NSAIDs was carried out at multiple wavelengths i.e., 281, 255, 271 and 219 nm for INP, KTP, IBP and FNP respectively.

## Synthesis

### Preparation of sporopollenin containing active site (Sp-TDI) (1)

2.0 g of sporopollenin and 10 mL of toluene-2,4-diisocyanate (TDI) were mixed with the help of a magnetic stirrer in 20 mL dry toluene. The functionalization was performed under nitrogen gas at room temperature for 4 hrs. Then, the resultant Sp-TDI was separated by centrifugation at 2,000 rcf for 5 min and sequentially washed with dry toluene. The sample was dried and stored in a desiccator.

### Preparation of p-tert-butylcalix[4]arene based sporopollenin (Sp-TDI-calix) (2)

0.411 g of *p*-tert-butylcalix[4]arene was dissolved in 20 mL of dry N,N-dimethylformamide (DMF) followed by the addition of 1.5 g of Sp-TDI (*1*) and stirred at 70 °C for 2 hrs using a magnetic stirrer in the presence of nitrogen gas. The resultant compound was washed with dichloromethane (DCM) and deionized water for the removal of unnecessary particles

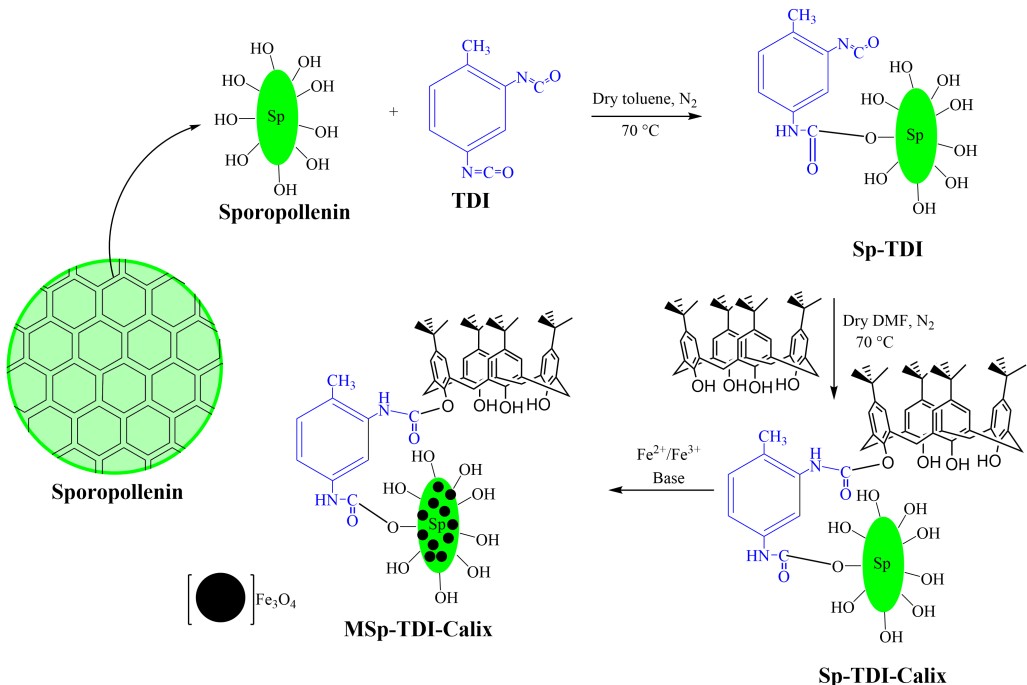

**Figure 2** **The synthesis pathways of MSp-TDI-calix adsorbent.**

absorbed on the sample surface. The sample was dried and stored in a desiccator and labelled as Sp-TDI-calix (*2*).

### *Preparation of magnetic p-tert-butylcalix[4]arene based sporopollenin (MSp-TDI-calix) (3)*

The magnetization of *p*-tert-butylcalix[4]arene functionalized TDI modified sporopollenin (Sp-TDI-calix) (*3*) was carried out as follows; 13.3 g of FeCl$_3$.6H$_2$O, 19.9 g of FeCl$_2$.4H$_2$O, 5 mL of HCl (5 M), 40 mL of deionized water and 5 mL of ethanol were mixed in a flask followed by heating to 40 °C until complete dissolution of the salts occured. Then, 1.0 g of Sp-TDI-calix (*2*) was re-dispersed in 30 mL of the solution and stirred for 2 h at room temperature. The Sp-TDI-calix (*3*) suspension was filtered and the filtrate was quickly washed with deionized water and immediately transferred into 0.1 M ammonia solution. After 2 h stirring at room temperature, the resultant compound was separated from the solution using an external magnet, washed thoroughly with deionized water and dried under vacuum. Fig. 2 illustrates the synthesis pathway of the MSp-TDI-calix (*3*) sorbent.

### Magnetic solid phase extraction (MSPE)

For the MSPE technique, 30 mg of MSp-TDI-calix (*3*) sorbent was added to 200 mL of spiked sample which contained a mixture of 1 μg/L INP, KTP, IBP and FNP respectively in deionized water at pH 4. The solution was horizontally shaken using a shaker for 30 min to allow the sorbent to disperse uniformly in the solution. Then, neodymium (Nd) magnet was used to isolate the sorbent from the solution. The solution became limpid after a few minutes and the upper solution was decanted. Afterwards, 1.5 mL of acetonitrile was added

to elute the NSAIDs adsorbed on MSp-TDI-calix (*3*) and horizontally shaken for another 30 min. The collected eluate was dried using a stream of nitrogen gas and re-dissolved in 0.7 mL of acetonitrile. Finally, 10 μL portion of the eluate was injected into HPLC for analysis.

### Real sample preparation

Different types of water samples i.e., tap water, drinking water and river water were utilized to check the matrix effects towards extraction of NSAIDs. Tap water was obtained from the analytical chemistry laboratory at the University of Malaya, Malaysia. Drinking water was purchased from a local store. River water was collected from Sungai Sendat, Selangor, Malaysia. The samples were stored at 4 °C prior to use.

## RESULTS AND DISCUSSIONS

### Characterization of sorbent

The chemical structure of the synthesized sorbent was characterized using FT-IR, XRD, FESEM-EDX, BET and VSM analyses. Figure 3 depicts the IR of Sp-TDI, Sp-TDI-calix and MSp-TDI-calix. In Sp-TDI, the presence of IR bands at 2,280 cm$^{-1}$ may be attributed to the isocyanate peak while appearance of the IR band at 1,645 cm$^{-1}$ confirmed the existence of aromatic rings. The IR bands at 1,604 cm$^{-1}$ and 1,233 cm$^{-1}$ indicates the presence of C=O and C–N stretches of the formed carbamate linkages between isocyanate proving the incorporation of isocyanate functionalities on the surface of sporopollenin. Meanwhile, the disappearance of reserved isocyanate peak in Sp-TDI-calix is due to the functionalization between Sp-TDI surface with *p*-tert-butylcalix[4]arene. Hence, the absence of the peak at 2,280 cm$^{-1}$ indicates that *p*-tert-butylcalix[4]arene was successfully bonded onto the surface of the Sp-TDI. Moreover, the appearance of bands at 1,677 cm$^{-1}$, 1,454 cm$^{-1}$ and 1,200 cm$^{-1}$ correspond to $C_{Ar}$–$C_{Ar}$ stretching, methylene bridge –CH$_2$- and C-C stretching respectively proving that *p*-tert-butylcalix[4]arene has been grafted on the surface of sporopollenin. Finally, IR bands at 566 cm$^{-1}$ are associated with the magnetite Fe$_3$-O$_4$ band in MSp-TDI-calix which indicates that the sorbent was successfully magnetized with MNPs.

The surface morphology of the sorbent was analysed with FESEM. The FESEM images of raw sporopollenin, Sp-TDI-calix and MSp-TDI-calix are depicted in Fig. 4. The image of raw sporopollenin in Fig. 4A shows a smooth morphology with an open and uniform interconnected hexagonal hollowed structure surface. Meanwhile, the FESEM image for Sp-TDI-calix in Fig. 4B shows the pores of sporopollenin changed to rough and lumpy filled with *p*-tert-butylcalix[4]arene particles. Figure 4C clearly shows that Fe$_3$O$_4$ particles were filled inside the hollow surface of sporopollenin and each hollow structure remained unchanged after the magnetization process. Hence, these images reveal that the functionalization process of *p*-tert-butylcalix[4]arene onto the surface of sporopollenin was successful and magnetite has also embedded inside the pores of sporopollenin.

Energy dispersive X-ray spectroscopy (EDX) analysis gave promising results in terms of elemental composition. Figures 5A and 5B represent the FESEM image and EDX spectra with the elemental composition of Sp-TDI-calix and MSp-TDI-calix respectively. As stated

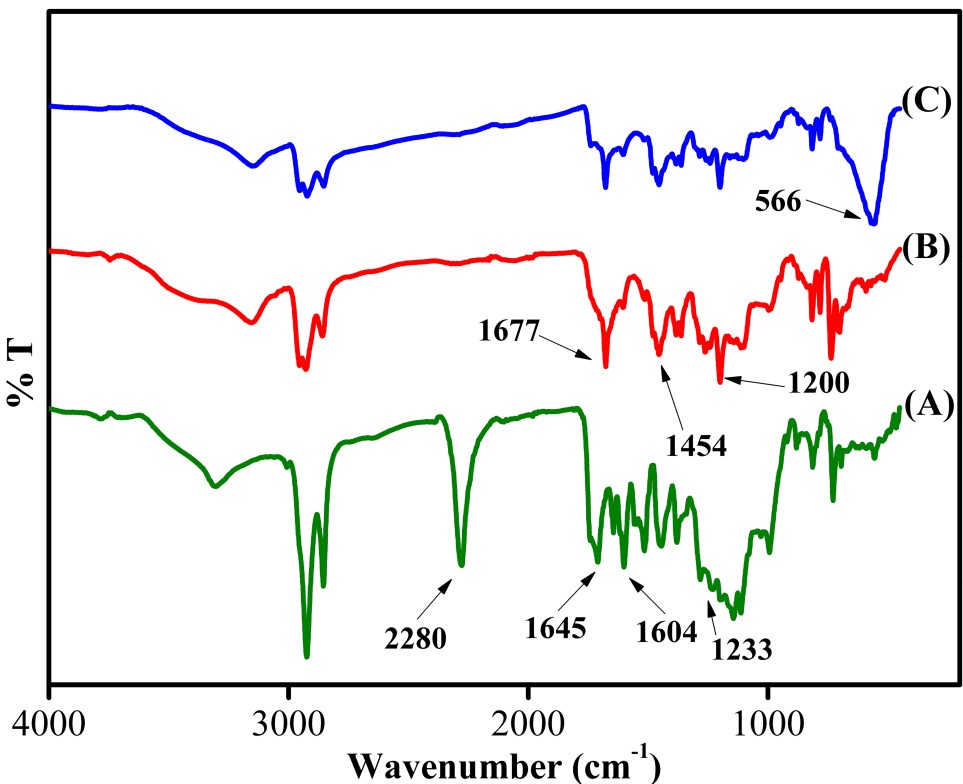

**Figure 3  FTIR spectra of synthesized adsorbent.** FTIR spectra of (A) Sp-TDI, (B) Sp-TDI-calix and (C) MSp-TDI-calix.

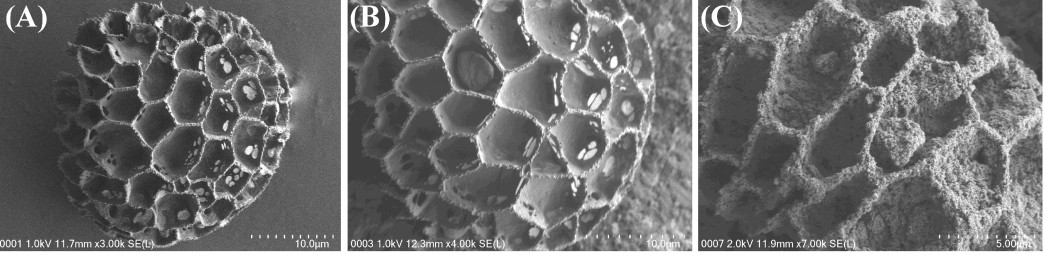

**Figure 4  FESEM images of synthesized adsorbent.** FESEM images of (A) sporopollenin, (B) Sp-TDI-calix and (C) MSp-TDI-calix.

earlier, raw sporopollenin only contains hydrogen, carbon and oxygen. The EDX spectrum of Sp-TDI-calix showed 14.4% nitrogen which originates from the isocyanate group after immobilization with *p*-tert-butylcalix[4]arene. Meanwhile, after the magnetization process with $Fe_3O_4$, EDX detected 33.8% iron present in MSp-TDI-calix indicating that the final material MSp-TDI-calix was successfully filled with MNP.

The XRD analyses were conducted to determine the crystallinity of Sp-TDI, Sp-TDI-calix, and MSp-TDI-calix. As represented in Fig. S1A, Sp-TDI exhibits broad diffraction

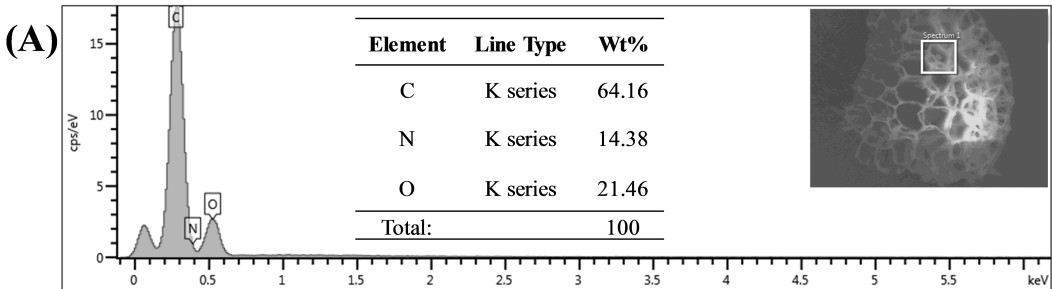

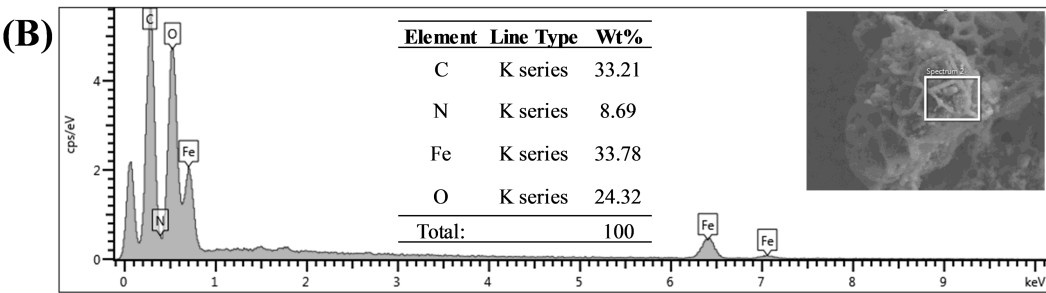

**Figure 5** **EDX spectra of synthesized adsorbent.** EDX spectra of (A) Sp-TDI-calix and (B) MSp-TDI-calix.

peaks at ∼25° which are typically observed for amorphous materials. Meanwhile, a few diffraction peaks appeared at ∼10°–30° when *p*-tert-butylcalix[4]arene was functionalized on the surface of Sp-TDI. The additional peaks, as displayed in Fig. S1B indicate that *p*-tert-butylcalix[4]arene improved the crystallinity of Sp-TDI-calix. In conformity with the Joint Committee on Powder Diffraction Standards (JCPDS) reference pattern of magnetite (00-019-0629), six diffraction peaks of $Fe_3O_4$ were observed in MSp-TDI-calix in Fig. S1C which are 220, 311, 400, 422, 511, and 440. These peaks are related to the cubic spine plane of $Fe_3O_4$ and thus confirmed the presence of $Fe_3O_4$. It is also observed that no distinct diffraction peaks of *p*-tert-butylcalix[4]arene appeared for MSp-TDI-calix which indicate that the *p*-tert-butylcalix[4]arene molecules were homogeneously distributed.

VSM technique was used to measure the strength of magnetization behaviour of MSp-TDI-calix. As shown in Fig. S2, MSp-TDI-calix sample displayed typical superparamagnetic behaviour. The saturation magnetization of the sample was 19.4 emu/g and sufficient for magnetic separation since the reported minimal magnetic strength for separation is 16.3 emu/g (*Cho et al., 2015*). This data proved that the prepared sorbent could be dispersed in water and separated by using neodymium magnet. The inset photos in Fig. S2 show the sorbent separation using external magnetic field.

Figure S3 shows the nitrogen adsorption–desorption isotherm of the sorbent MSp-TDI-calix. It exhibits close to type IV as defined by the IUPAC classification with an evident hysteresis loop in the range of 0.05–1.0 and suggests that the sorbent is basically a mesoporous material (*Farhadi, Safabakhsh & Zaringhadam, 2013*). The specific surface area of the sample calculated by the BET is 26.5 m² g⁻¹. Moreover, the pore size can be calculated

from the surface area according to the equation $4V/S_{BET}$, where V is adsorption total pore volume and $S_{BET}$ is the specific surface area of the MSp-TDI-calix. The pore size calculated from the surface area data is approximately 18.8 nm. Table S1 summarizes the pore size, pore volume and $S_{BET}$ value for sporopollenin, Sp-TDI-calix and MSp-TDI-calix. The lower BET surface area in Sp-TDI-calix in comparison to raw sporopollenin from 2.24 m$^2$/g–2.08 m$^2$/g could be due to the adsorption sites being covered by *p*-tert-butylcalix[4]arene which have immobilized on the surface of Sp-TDI, hindering the N$_2$ molecules from accessing the binding site. Thus, it showed that the *p*-tert-butylcalix[4]arene was successfully bonded to the Sp-TDI surface (*Huang et al., 2010*). The surface area, pore volume and pore size of the sorbent increased due to dispersity of the particles, and therefore, enhanced the adsorption capacity especially for large adsorbate molecules (*Fan et al., 2011*).

## Optimization of MSPE

Herein, the prepared bio-polymer sorbent MSp-TDI-calix were used as MSPE sorbent to evaluate the extraction of target analytes namely INP, KTP, IBP and FNP from the NSAIDs group. In order to achieve optimal conditions for the MSPE procedure, various parameters including sorbent amount, extraction time, sample volume, type of organic eluent, volume of organic eluent, desorption time and sample solution pH were investigated.

### Sorbent amount

Sorbent amount is the most important parameter to improve the recovery of target analytes. Different amounts of sorbent ranging from 5 to 50 mg were evaluated to determine the extraction efficiency. As shown in Fig. 6A, peak area increased with increasing amounts of sorbent up to 30 mg due to availability of accessible sites for interaction. Increasing the sorbent amount increases the adsorption active sites and thus leads to higher peak areas (*Rashidi Nodeh et al., 2017*). However, above 30 mg sorbent amount, the peak area showed a slightly decreasing trend because of the weak elution by a fixed volume of the eluent towards the adsorbed analytes (*Tahmasebi & Yamini, 2014*). The fixed volume of the eluent solvent was insufficient to achieve a complete dissolution of the sorbent (*Arghavani-Beydokhti, Rajabi & Asghari, 2018*). The larger sorbent amount at definite volume and specific contact time possibly decreased the elution efficiency and thus reduced the peak area obtained (*Shahriman et al., 2018*). Moreover, the loss of sorbent amount during the separation process by an external magnet may have also lead to a decrease in peak areas (*Bagheri, Roostaie & Baktash, 2014*). Thus, 30 mg was chosen to extract targeted NSAIDs from aqueous samples for all subsequent experiments.

### Sample volume

Sample volume plays an important role in recovering the target analytes. High enrichment factor and high sensitivity can be obtained by using a large sample volume (*Soutoudehnia Korrani et al., 2016*). Thus, six different volumes of water samples ranging from 20 to 250 mL were studied. As shown in Fig. 6B, the highest peak area was obtained with 200 mL sample volume. However, beyond 200 mL sample volume, the peak area decreased for KTP especially probably due to the breakthrough volume being exceeded (*Abd Rahim et al., 2016*). The recovery of INP, IBP and FNP did not increase significantly when 250 mL

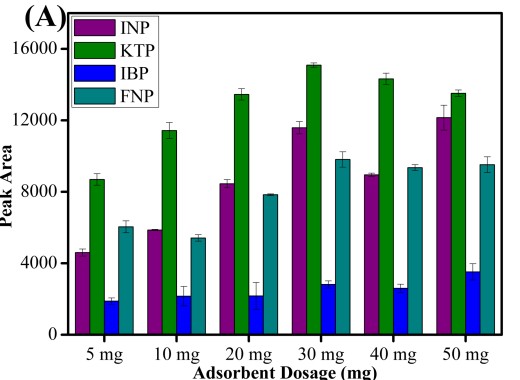
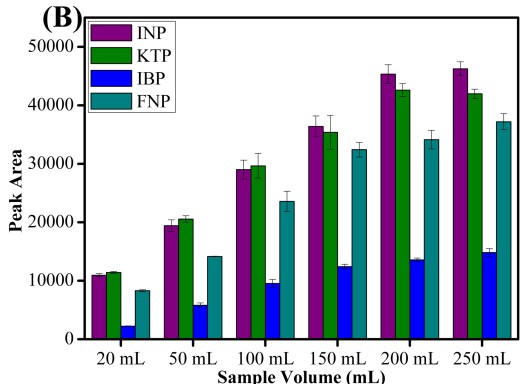

**Figure 6 The effect of adsorbent dosage and sample volume.** The effect of (A) adsorbent dosage, and (B) sample volume for the extraction of NSAIDs using MSp-TDI-calix and analysis using HPLC-DAD. HPLC conditions: acidified (1% with acetic acid) water/acetonitrile (50:50 v/v) as a mobile phase at a flow rate of 1 mL min$^{-1}$, the HPLC column temperature was set at 40 °C, the sample injection volume was 10 μL, the DAD detection for the selected NSAIDs was carried out at multiple wavelengths i.e., 281, 255, 271 and 219 nm for INP, KTP, IBP and FNP respectively.

sample loading volume was used. Therefore, 200 mL of sample volume was selected as optimum volume for further extraction of NSAIDs in water samples.

### Extraction and desorption time

The extraction and desorption time profiles was obtained by varying the time between 5–60 min. Based on the results shown in Fig. 7A, it is evident that the maximum peak area was obtained at an extraction time of 30 min and a further increase in extraction time lowered the extraction recovery. Meanwhile, for the desorption process, 10 min was adequate to elute all target analytes from the sorbent surface as shown in Fig. 7B. Since MSPE is a dynamic process, after 30 min of extraction process and desorption the analytes may be destroyed when the process period is prolonged (*Xue, Li & Xu, 2017*). Hence, the extraction and desorption efficiency decreased at longer time periods. It can be concluded that 30 min of extraction and 10 min of desorption time were sufficient for further experiments.

### Type of organic eluent and volume

Seven types of organic solvents, namely methanol, acetonitrile, *n*-hexane, toluene, chloroform, ethyl acetate, dichloromethane and dimethylformamide were used as desorption solvents to examine their role towards the extraction efficiencies of studied NSAIDs. As shown in Fig. 8A, acetonitrile showed the best result to desorb NSAIDs with an overall high recovery compared to other organic solvents. These phenomena can be explained by the molecular interaction between the analytes and the sorbent surface. The intermolecular forces that may occur between MSp-TDI-calix and NSAIDs are hydrogen bonding, dipole–dipole interaction, $\pi-\pi$ interaction and van der waals forces. The organic eluent disrupts the retentive intermolecular forces between sorbent surface and analyte. Methanol is expected to give a strong eluent effect on polar polymeric sorbents, but acetonitrile gives excellent eluent effect on sorbent surface compared with methanol. This

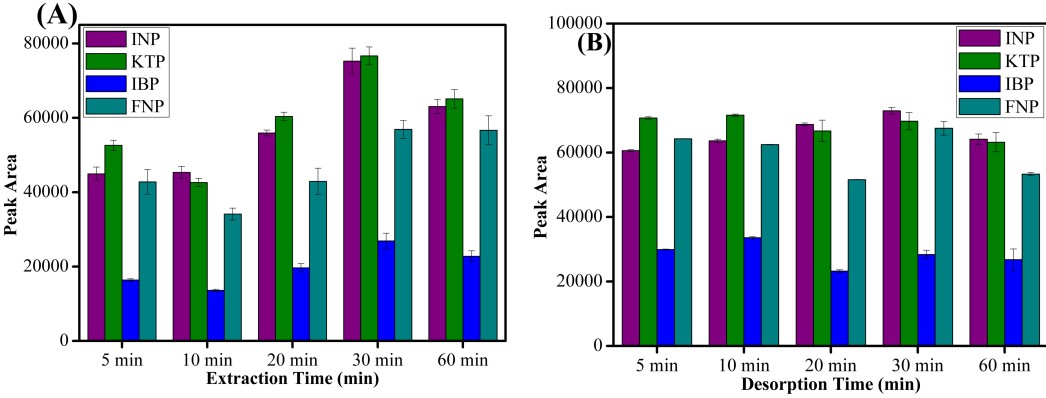

**Figure 7  The effect of extraction time and desorption time.** The effect of (A) extraction time, and (B) desorption time on the extraction of NSAIDs from 5–60 min using MSp-TDI-calix and analysis using HPLC-DAD. HPLC conditions: acidified (1% with acetic acid) water/acetonitrile (50:50 v/v) as a mobile phase at a flow rate of 1 mL min$^{-1}$, the HPLC column temperature was set at 40 °C, the sample injection volume was 10 $\mu$L, the DAD detection for the selected NSAIDs was carried out at multiple wavelengths i.e., 281, 255, 271 and 219 nm for INP, KTP, IBP and FNP respectively.

is because methanol has a strong polar eluent strength and is unable to disrupt non-polar interaction sites. Acetonitrile has mid to polar-apolar eluent strength that can disrupt the binding mechanism at polar and non-polar site sorbent surfaces. Nonpolar solvents such as *n*-hexane have poor eluent capability due to their non-polar characteristic to disrupt the polar interaction. Effects due to volume of the organic eluent were also determined. As presented in Fig. 8B, six different volumes ranging from 0.5 to 3.0 mL of acetonitrile were optimized. The maximum peak area was observed with 1.5 mL volume of acetonitrile. In contrast, above 1.5 mL of acetonitrile, the peak area decreased due to dilution factor and loss of analytes to environment at higher volume of organic eluent (*Shukri et al., 2015*; *Hu et al., 2017*; *Rashidi Nodeh et al., 2017*). During drying the elution extract by nitrogen gas stream, the higher volume needed more time to dry and preconcentrate the analytes. Thus, some of analytes may be lost to the surroundings during this period and caused a decrease in peak area. Therefore, 1.5 mL of acetonitrile was selected to be utilized in further experiments.

### *Solution pH*

The pH condition plays an important role on the extraction efficiency of analytes as changes in pH can influence the charges on both the sorbent and analytes. As clearly shown in Fig. 9, the extraction of NSAIDs is pH dependent. It is obvious that the peak area increased from pH 2 to pH 4 and decreased as the pH value increased further from pH 6 to pH 12. This observation is related to the pK$_a$ value of the selected NSAIDs. Most of the NSAIDs have a pK$_a$ value in the range of 3.66–4.88. At pH < pK$_a$ value, NSAIDs mostly exist in their neutral form, which leads to the formation of intermolecular forces such as hydrogen bonding, $\pi$–$\pi$ interaction and hydrophobic interaction between MSp-TDI-calix and the target NSAIDs. Therefore, highest extraction efficiency was achieved below pH 4.88. In the case whereby pH > pK$_a$ value, ionization process takes place and the NSAIDs transform

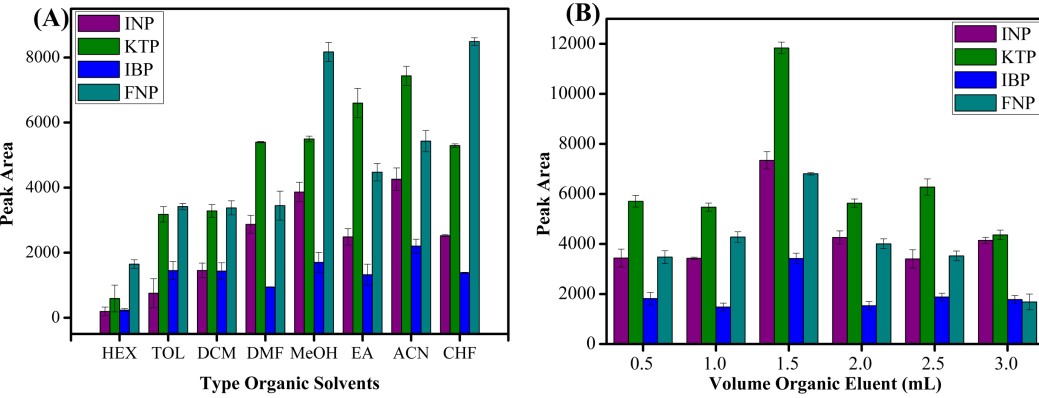

**Figure 8** **The effect of type of organic eluent and volume of organic eluent.** The effect of (A) type of organic eluent, and (B) volume of organic eluent for the extraction of NSAIDs using MSp-TDI-calix and analysis using HPLC-DAD. HPLC conditions: acidified (1% with acetic acid) water/acetonitrile (50:50 v/v) as a mobile phase at a flow rate of 1 mL min$^{-1}$, the HPLC column temperature was set at 40 °C, the sample injection volume was 10 µL, the DAD detection for the selected NSAIDs was carried out at multiple wavelengths i.e., 281, 255, 271 and 219 nm for INP, KTP, IBP and FNP respectively.

to their anion form, severely weakening the existing intermolecular forces (*Luo et al., 2011*; *Fan et al., 2014*). As a result, the extraction efficiency of target analytes decreased. Therefore, pH 4.0 was selected as the optimum pH for subsequent analysis of NSAIDs. In addition, the designed nanocomposite performing well under acidic conditions also indicates the suitability of sporopollenin as a solid support for MNPs in overcoming the instability of the naked MNPs in acidic medium.

### Reusability of the sorbent

Recycling of sorbents is crucial for practical applications to improve cost effectiveness. Figure S4 displays the efficiency of MSp-TDI-calix sorbent for extraction of selected NSAIDs when regenerated up to five cycles. The sorbent was washed with acetonitrile before proceeding to subsequent MSPE applications. The recovery efficiency showed stability even after five extractions, with 80% efficiency from the first cycle. Thus, the MSp-TDI-calix magnetic particles are mechanically stable and possess good reusability.

## Method validation

Optimal conditions for extraction of selected NSAIDs using the proposed MSPE method were 30 mg sorbent amount, 200 mL sample volume, 30 min extraction and 10 min desorption time, 1.5 mL acetonitrile as organic eluent and pH 4 of sample solution. Linearity, LOD, LOQ and precision (intra-day and inter-day) were evaluated to validate the studied method under these optimal conditions. As listed in Table 1, all tested NSAIDs showed good linearity with good coefficient of determination ($R^2 \geq 0.991$). LOD was calculated based on signal-to-noise ratio ($3 \times$ SD/$m$) ($n = 10$) and the values obtained were in the range of 0.07 to 0.26 µg/L. Meanwhile, LOQ with signal-to-noise ratio ($10 \times$ SD/$m$) ($n = 10$) was also evaluated and the values obtained were in the range of 0.2–0.89 µg/L. Precision was studied in terms of repeatability (intra-day), ($n = 5$) and reproducibility

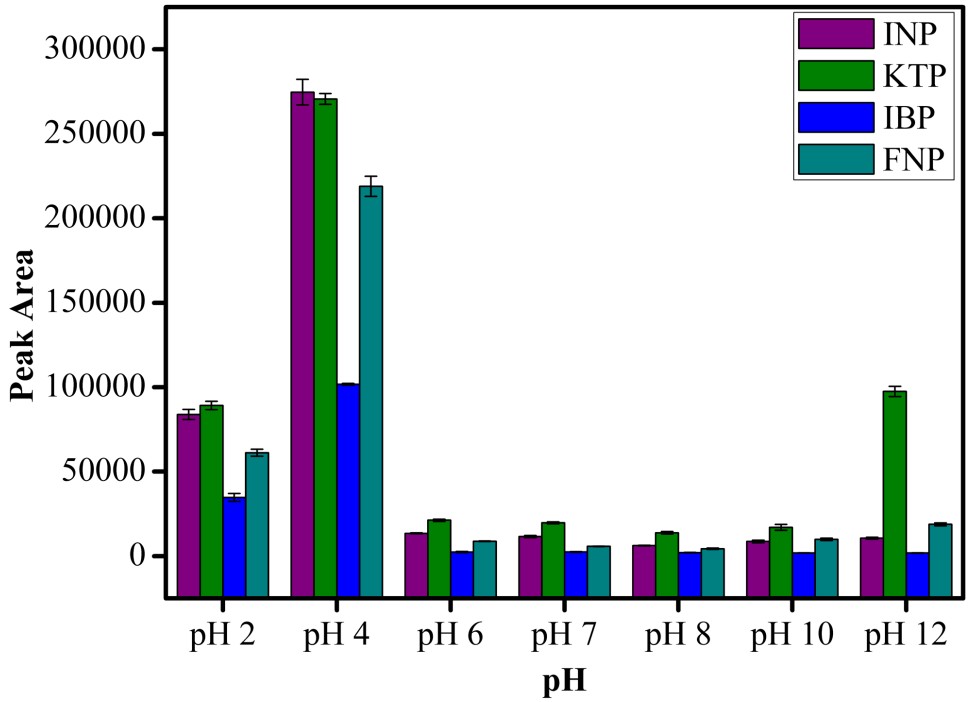

**Figure 9** **The effect of solution pH for the extraction of NSAIDs using MSp-TDI-calix and analysis using HPLC-DAD.** HPLC conditions: acidified (1% with acetic acid) water/acetonitrile (50:50 v/v) as a mobile phase at a flow rate of 1 mL min$^{-1}$, the HPLC column temperature was set at 40 °C, the sample injection volume was 10 μL, the DAD detection for the selected NSAIDs was carried out at multiple wavelengths i.e., 281, 255, 271 and 219 nm for INP, KTP, IBP and FNP respectively.

**Table 1** **Qualitative data of the proposed MSPE technique.**

| Analyte | Linearity (μg/L) | Regression equation | $R^2$ | LOD (μg/L) | LOQ (μg/L) | Precision | |
|---|---|---|---|---|---|---|---|
| | | | | | | Intra (RSD% $n=5$) | Inter (RSD% $n=15$) |
| INP | 0.5–500 | $y = 219677x + 3773$ | 0.9930 | 0.074 | 0.25 | 2.4 | 2.5 |
| KTP | 0.5–500 | $y = 213367x + 4440$ | 0.9916 | 0.061 | 0.20 | 3.2 | 2.8 |
| IBP | 0.5–500 | $y = 58351x + 851$ | 0.9951 | 0.267 | 0.89 | 3.9 | 3.1 |
| FNP | 0.5–500 | $y = 170853x + 1283$ | 0.9993 | 0.111 | 0.37 | 2.4 | 3.2 |

(inter-day), ($n = 15$) and expressed as relative standard deviation (RSD). The intra-day precision was demonstrated by performing five replicates of standard solutions on a single day. The inter-day precision of the MSPE procedure was evaluated by performing standard solutions of the same concentration levels in five replicates on three consecutive days. The results showed that the intra-day and inter-day RSDs were in the range of 2.4–3.9% and 2.5–3.2% respectively. Results obtained for the LOD, LOQ and repeatability precision as shown in Table 1 indicate that the method had high sensitivity and repeatability.

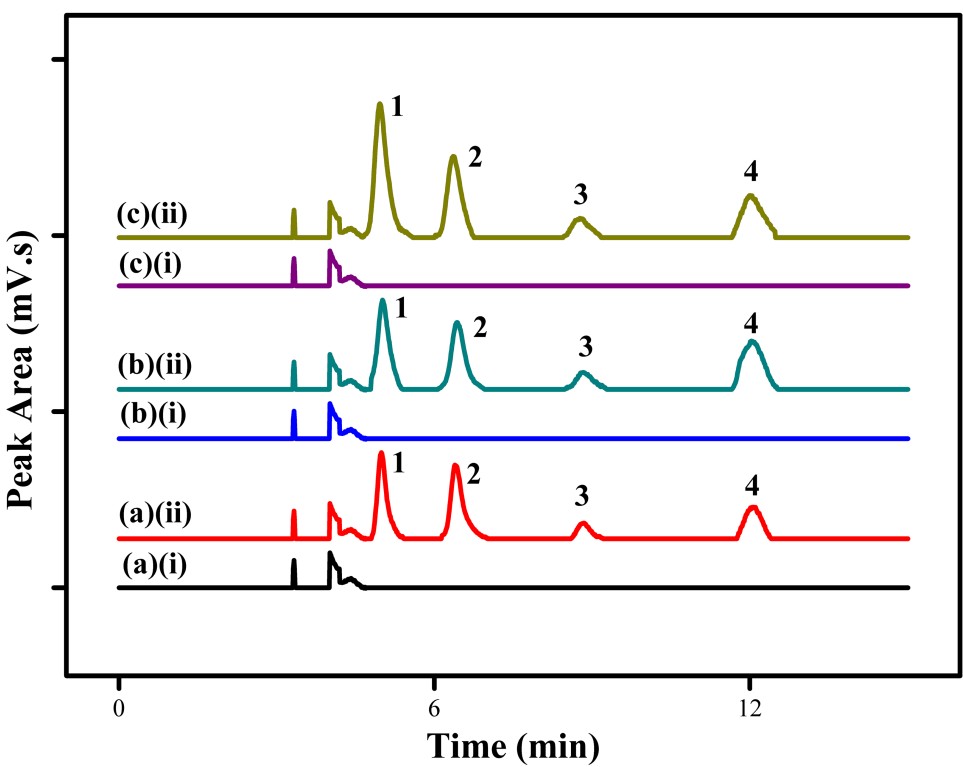

**Figure 10  HPLC chromatograms of water samples.** HPLC chromatograms of water samples (spiked with 100 ug/L of each NSAIDs) using the proposed MSp-TDI-calix MSPE method; (A)(i) non-spiked tap water; (A)(ii) spiked tap water; (B)(i) non-spiked drinking water; (B)(ii) spiked drinking water; (C)(i) non-spiked river water; (C)(ii) spiked river water. Peak identification: (*1*) INP, (*2*) KTP, (*3*) IBP, and (*4*) FNP.

## Environmental water samples analysis

The proposed MSPE method using MSp-TDI-calix sorbent for extraction of selected NSAIDs was applied to the analysis of three different types of environmental water samples (tap water, drinking water and river water). In order to determine the effect of sample matrix, the real samples were spiked with four NSAIDs: INP, KTP, IBP and FNP at the concentration of 10 µg/L and 100 µg/L under optimized conditions. Figure 10 displays the chromatogram of spiked water samples containing 10 µg/L of NSAIDs compared with non-spiked samples. The results which are tabulated in Table 2 show satisfactory recovery of NSAIDs in tap water ranging from 88.1%–110.7% with RSDs ($n = 5$) in the range of 1.6%–4.6%. For drinking water, the recovery achieved range between 91.9%–107.9% with RSDs ($n = 5$) from 1.9%–4.6%. Meanwhile, NSAIDs recovery in river water samples showed excellent recovery in the range of 94.3%–115.8% with RSDs ($n = 5$) from 1.8%–4.5%. Based on the results obtained, the good recoveries of the selected NSAIDs in the three different sample matrices at trace levels shows the high potential of this MSPE technique. However, no analyte in the non-spiked real samples were detected.

**Table 2**  Percentage relative recovery and RSD ($n = 5$) of NSAIDs in spiked water samples extracted with MSp-TDI-calix.

| Analyte | Spiked (ng/mL) | Tap water ($n = 5$) | | Drinking water ($n = 5$) | | River water ($n = 5$) | |
|---|---|---|---|---|---|---|---|
| | | Recovery (%) | RSD (%) | Recovery (%) | RSD (%) | Recovery (%) | RSD (%) |
| INP | 10 | 92.4 | 3.6 | 104.6 | 1.9 | 103.6 | 4.0 |
| | 100 | 99.6 | 4.1 | 107.9 | 4.0 | 103.3 | 1.8 |
| KTP | 10 | 88.1 | 2.1 | 107.8 | 2.1 | 98.2 | 2.2 |
| | 100 | 97.4 | 1.6 | 105.7 | 4.6 | 106.5 | 3.2 |
| IBP | 10 | 109.8 | 4.6 | 105.2 | 3.0 | 94.3 | 4.5 |
| | 100 | 110.7 | 1.8 | 94.3 | 4.3 | 115.8 | 1.9 |
| FNP | 10 | 103.5 | 4.4 | 96.2 | 2.6 | 97.2 | 3.5 |
| | 100 | 103.2 | 4.3 | 91.9 | 3.6 | 107.0 | 3.2 |

**Table 3**  Comparison of the current work to other reported MSPE techniques and sample matrices for determination of NSAIDs.

| Matrix | Adsorbent | Technique | Linearity range (ng/mL) | Recovery (%) | LODs (ng/mL) | Ref |
|---|---|---|---|---|---|---|
| Blood | $Fe_3O_4$@$SiO_2$@IL | MSPE/HPLC UV | 0.5–100 | 92–97 | 0.2–0.5 | (*Amiri et al., 2016*) |
| Water | Metal organic framework $Fe_3O_4$/MIL-101(Cr) | MSPE/UHPLC-MS/MS | 0.1–25 | NR | 0.01–0.19 | (*Wang et al., 2017*) |
| Water and river water | $C_{18}$/Diol-$Fe_3O_4$ MNPs | MSPE/HPLC UV | 5–800 | NR | 0.42–1.44 | (*Luo et al., 2015*) |
| Urine And sewage | MNPs modified with cetyltrimethyl- ammonium bromide (CTAB) | MSPE/HPLC DAD | 7–200 | 91–97 | 2–7 | (*Khoeini Sharifabadi et al., 2014*) |
| Tap, drinking and river water | **MSp-TDI-calix** | **MSPE/HPLC DAD** | **5–500** | **88–116** | **0.06–0.26** | **Current study** |

**Notes.**
*NR-not reported

## Comparative study

The efficiency of the proposed MSp-TDI-calix was compared with similar MSPE methods reported in the literature. Table 3 summarizes several sorbents used for the determination of NSAIDs using MSPE method in different matrices in terms of linearity, LOD, and recovery. As can been seen, this work achieved excellent recovery, lower LODs as well as improved sensitivity compared with other reported methods. The LOD achieved by the present work is lower compared to other research studies using the MSPE method. These results demonstrate the great potential of the proposed method for analysis of NSAIDs in water samples.

## CONCLUSIONS

The extraction of NSAIDs on magnetic bio-polymer grafted with *p*-tert-butylcalix[4]arene (MSp-TDI-calix) utilizing the MSPE technique was studied. The synthesized material was successfully grafted and proved with various instruments including spectroscopy,

morphology, elementary, magnetic behaviour and thermal characterization analyses. The new sorbent in combination with HPLC-DAD enable sensitive analysis of selected NSAIDs in environment sample matrices such as tap water, drinking water and river water at low levels. The extraction of selected NSAIDs gave acceptable and low LOD values in the range of 0.06–0.26 µg/L and high recovery rates ranging from 88.1–115.8% for tested real samples. These advantages together with the effective regeneration and stability of the sorbent indicate its applicability to be employed as an alternative sorbent for enrichment and preconcentration of NSAIDs in environmental sample matrices.

## ACKNOWLEDGEMENTS

The authors are grateful to administrative authorities at the Department of Chemistry, UM. Special thanks go to Mrs. Ruhaida Bahru, the Sr. Asst. Science Officers and Mr. Nor Affandi Tamar, Asst. Science Officers at the Environmental Research Laboratory, for their kind support in the analysis work. Help rendered by the UM postgraduate students Siti Khalijah Mahmad Rozi and Ahmad Razali Ishak was invaluable.

### Funding

This work was supported by the Universiti Malaya Research Grant (Project No RP020A-16SUS and Project No RG381-17AFR) and Postgraduate Research Grant (Project No. PG046-2015A). The funders had no role in study design, data collection and analysis, decision to publish, or preparation of the manuscript.

### Grant Disclosures

The following grant information was disclosed by the authors:
Universiti Malaya Research Grant: RP020A-16SUS, RG381-17AFR.
Postgraduate Research Grant: PG046-2015A.

### Competing Interests

The authors declare there are no competing interests.

### Author Contributions

- Syed Fariq Fathullah Syed Yaacob conceived and designed the experiments, performed the experiments, analyzed the data, prepared figures and/or tables, authored or reviewed drafts of the paper, approved the final draft.
- Arniza Khairani Mohd Jamil prepared figures and/or tables, authored or reviewed drafts of the paper, approved the final draft.
- Muhammad Afzal Kamboh contributed reagents/materials/analysis tools, authored or reviewed drafts of the paper, approved the final draft.
- Wan Aini Wan Ibrahim contributed reagents/materials/analysis tools, authored or reviewed drafts of the paper, approved the final draft.
- Sharifah Mohamad conceived and designed the experiments, analyzed the data, contributed reagents/materials/analysis tools, authored or reviewed drafts of the paper, approved the final draft.

### Data Availability

Mohamad, Sharifah; Yaacob, Syed Fariq Fathullah Syed; Kamboh, Muhammad Afzal; Ibrahim, Wan Aini Wan; Jamil, Arniza Khairani Mohd (2018): Raw dataset.rar. figshare. Dataset. https://doi.org/10.6084/m9.figshare.5932720.v1.

### Supplemental Information

Supplemental information for this article can be found online at http://dx.doi.org/10.7717/peerj.5108#supplemental-information.

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
