# Peer review of "Fabrication of calixarene-grafted bio-polymeric magnetic composites for magnetic solid phase extraction of non-steroidal anti-inflammatory drugs in water samples"

_PeerJ, doi:10.7717/peerj.5108_

## Round 0.1 · original submission · Major Revisions

Initially, I was inclined to reject the manuscript. However, I believe that the substantive comments (from Reviewers 1 and 4) can be addressed. Please pay particular attention to these comments and revise the manuscript accordingly.

Reviewer 1 ·

Basic reporting

English should be revised by a native speaker throughout the manuscript. Some sentences are not well-written and are hard to understand.

Abbreviations should be revised. Some terms for which abbreviations have been provided appear latter in the text and should be replaced by their abbreviations.

Experimental design

In the work by Yaacob and co-workers, authors have synthetized a calixarene framework functionalized bio-polymeric magnetic composite and used it for the magnetic solid-phase extraction (m-dSPE) of four non-steroidal anti-inflammatory drugs from tap, drinking and river water samples prior their determination by HPLC-DAD. In particular, authors carried out a suitable characterization of the sorbent by FT-IR, XRD, FESEM, EDX, VSM and BET analysis. They also optimized parameters such as sorbent amount, extraction time, sample volume, type and volume of elution solvent, desorption time and pH of the sample in order to maximize the extraction efficiency of the procedure. Finally, the methodology was validated and applied to real sample analysis. However, and despite the interest of the use of this sorbent, some aspects, especially related with the optimization of the extraction procedure, preclude the publication of this manuscript in this journal:
1) During the optimization of the extraction, authors selected 30 minutes as the time which provided the highest peak areas. Although it seems to be justified by the data shown in Figure 7 (A), 30 minutes is a relatively long extraction time. Considering that the next value that is below is 20 minutes, authors should have done some tests between 20 and 30 minutes of extraction to check if peak areas decreased or not. They should have tried to reduce the extraction time.
2) The values of peak areas obtained with 10 minutes of elution are practically the same as those obtained with 5 minutes. Moreover, the repeatability obtained in both cases is also very similar. However, authors opted for 10 minutes.
3) Regarding the elution step, ACN was selected as the elution solvent which provided the highest peak areas. However, MeOH was the best solvent for the elution of fenoprofen and similar recoveries were found for another analyte in both cases. Thus, a mixture of both sorbents should have been tested. Authors commented that when more than 1.5 mL of ACN were used, the peak areas decreased due to the dilution factor. However, such decrease is not justified since in the description of the m-dSPE procedure they say that the elution extract was dried and then reconstituted in 0.7 mL of ACN. If that is so, the peak area should be practically the same using 1.5 mL or using higher volumes, considering that this volume is enough for the complete elution of the target analytes. On the contrary, the values obtained with 2 and 2.5 mL of ACN are very similar to the ones obtained with 0.5 and 1.0 mL.
4) Authors say that best peak areas are obtained at pH 4 and they try to justify it by means of the pKa of the analytes. However, considering the pKa values of these analytes, at pH 4 some of them will be ionized while others will be in their neutral form. Therefore, peak area values obtained at pH 4 are not justified. Further tests should be done at pH 3 and 5.
In conclusion, the optimization of the methodology was not properly developed, some experiments are missing, results are not properly justified and some of them are incongruous.

Validity of the findings

The bad experimental design make that the obtained results are not reliable.

Additional comments

In the work by Yaacob and co-workers, authors have synthetized a calixarene framework functionalized bio-polymeric magnetic composite and used it for the magnetic solid-phase extraction (m-dSPE) of four non-steroidal anti-inflammatory drugs from tap, drinking and river water samples prior their determination by HPLC-DAD. In particular, authors carried out a suitable characterization of the sorbent by FT-IR, XRD, FESEM, EDX, VSM and BET analysis. They also optimized parameters such as sorbent amount, extraction time, sample volume, type and volume of elution solvent, desorption time and pH of the sample in order to maximize the extraction efficiency of the procedure. Finally, the methodology was validated and applied to real sample analysis. However, and despite the interest of the use of this sorbent, some aspects, especially related with the optimization of the extraction procedure, preclude the publication of this manuscript in this journal:
1) During the optimization of the extraction, authors selected 30 minutes as the time which provided the highest peak areas. Although it seems to be justified by the data shown in Figure 7 (A), 30 minutes is a relatively long extraction time. Considering that the next value that is below is 20 minutes, authors should have done some tests between 20 and 30 minutes of extraction to check if peak areas decreased or not. They should have tried to reduce the extraction time.
2) The values of peak areas obtained with 10 minutes of elution are practically the same as those obtained with 5 minutes. Moreover, the repeatability obtained in both cases is also very similar. However, authors opted for 10 minutes.
3) Regarding the elution step, ACN was selected as the elution solvent which provided the highest peak areas. However, MeOH was the best solvent for the elution of fenoprofen and similar recoveries were found for another analyte in both cases. Thus, a mixture of both sorbents should have been tested. Authors commented that when more than 1.5 mL of ACN were used, the peak areas decreased due to the dilution factor. However, such decrease is not justified since in the description of the m-dSPE procedure they say that the elution extract was dried and then reconstituted in 0.7 mL of ACN. If that is so, the peak area should be practically the same using 1.5 mL or using higher volumes, considering that this volume is enough for the complete elution of the target analytes. On the contrary, the values obtained with 2 and 2.5 mL of ACN are very similar to the ones obtained with 0.5 and 1.0 mL.
4) Authors say that best peak areas are obtained at pH 4 and they try to justify it by means of the pKa of the analytes. However, considering the pKa values of these analytes, at pH 4 some of them will be ionized while others will be in their neutral form. Therefore, peak area values obtained at pH 4 are not justified. Further tests should be done at pH 3 and 5.
In conclusion, the optimization of the methodology was not properly developed, some experiments are missing, results are not properly justified and some of them are incongruous.
Other specific comments:
- Authors selected 30 mg of sorbent as the amount which provided the highest peak areas. However, the explanation they gave to justify the decreasing in peak areas when higher amounts of sorbent were used is not clear. Authors should rewrite this or find a suitable explanation to this fact.
- English should be revised by a native speaker throughout the manuscript. Some sentences are not well-written and are hard to understand.
- Abbreviations should be revised. Some terms for which abbreviations have been provided appear latter in the text and should be replaced by their abbreviations.
- Were LODs and LOQs experimentally checked?
- Authors should explain how the mixture of the sample and the sorbent was carried out during the extraction. How did they stir the mixture?
- How was the calibration curve experimentally developed?
- Details about organic solvents, mobile phase additives, analyte standards, etc. should be given in the materials section.
- ‘Sorbent amount’ should be used instead of ‘adsorbent dosage’.
- Concentration units should be changed to ‘μg/L’, since ‘mL’ is not an International System unit.
- Complete equations of the calibration curves should be provided in Table 1.

Reviewer 2 ·

Basic reporting

It’s highly significant for public health to monitor medicinal compounds in aquatic environments. This work reports on an MSPE procedure for analysis of NSAIDs in water using a novel magnetic nano-composite material. The study is well designed and performed. Results are solid. The proposed MSPE-HPLC-UV method has advantages over the existing methods. It’s my recommendation that this work is accepted for publication with minor revisions.

Experimental design

Study is well designed .

Validity of the findings

The results are solid.

Additional comments

1. Solution pH has a huge impact on the MSPE efficiency. However, it’s not mentioned through out the article how the pH was maintained at 4.
2. Caption for Fig 10. The spiking concentrations of NSAIDs should be stated here.

Reviewer 3 ·

Basic reporting

This work described the fabrication and characterization of a material for MSPE, and its performance for extracting several pharmaceuticals were assessed.
The paper was written in a logical order, with clear language. Literature review was comprehensive, and sufficient background was provided in the introduction.
The structure of this paper is well organized, with raw data shared in supporting information. So in general, I think this work meet the standard of PeerJ, and should be accepted for publication after minor revisions.

Experimental design

The research question was relevant and clearly defined. Well controlled experiments were conducted, and methods were described with enough details that readers should be able to replicate. From the information provided, the investigation was performed to a high standard.

Validity of the findings

Most of the data look solid and convincing, and the figures and tables support the conclusions. The conclusions are well stated, and
The only two questions I have on the results are:
1, the use of peak areas as a measure of analytes concentration, in Figure 6,7,8,9 and SI 4. I understand that all the recoveries are based on relative comparison, but why not quantify the concentration in accurate units?
2, the authors mentioned recoveries in the text, but I did not see the baseline (100%) in any of those bar plots. This cause some difficulties when connecting the text to the figures. Standard bars or a horizontal lines should present in those figures.

Additional comments

Line 56, "These drugs pose toxic effects in aquatic ecosystems and may cause harm to not only marine life but also to humans"you will need reference for such statement.
Figure 1, I see there is some notice about "Auto Gamma Correction" used, which seems to blur the figure. After all, these are common pharmaceuticals, why not just use plain presentation for the structures (like those in Figure 2)?

Reviewer 4 ·

Basic reporting

A more sufficient survey of background should be performed, please focus on the synthesis of calixarene modified magnetic nanoparticles. The relative information should be added in Introduction.

Experimental design

Overall, the experimental design of this manuscript was quite thorough, except that the adsorption capacity of the synthesized nano composite should be better added.

Validity of the findings

No comment.

Additional comments

The author synthesized calixarene-grafted biopolymeric magnetic nanocomposite, and used it as the adsorption medium for the analysis of nonsteroidal anti-inflammatory drugs in environmental water. The nanocomposite was carefully characterized, the extraction conditions were thoroughly evaluated, and compared to previously reported methods, the proposed method exhibited a better performance. However, there are some issues need to be addressed before publication. Therefore, I suggestion a major revision of this manuscript should be made before the acceptance by Peer J. The specific comments are listed as follows.
1. In the Introduction, the instable property of naked MNPs in acidic environment or other harsh condition were indicated as a disadvantage. However, Fe3O4 was immobilized on the surface of the designed nanocomposite, thus directly exposed to acidic solution when extracting samples. It seems controversial. Please revise the manuscript to make the points consistently.
2. Many calixarene modified magnetic nanoparticles were reported so far. Please compare the advantages of the prepared nanoparticles with the reported ones in the aspect of synthetic methods, etc.
3. Please indicate the binding mechanism of caixarene with guest molecules more clearly. Are there other interactions existed aside from pai-pai stacking interaction, hydrophobic or hydrophilic interaction? In other words, why can NSAIDs be selectively extracted by caixarene modified nanocomposite?
4. An adsorption isotherm and adsorption capacity should be investigated to elucidate the adsorption behavior between sorbent and NSAID.
5. It would be better if Figures 6-9 could be merged into one.

---

## Round 0.2 · accepted · Accept

Thank you for your efforts in addressing reviewer comments and making revisions to the manuscript. I have checked the revisions and feel that the concerns raised by the reviewers were addressed.

#